# In Silico Pleiotropy Analysis in KEGG Signaling Networks Using a Boolean Network Model

**DOI:** 10.3390/biom12081139

**Published:** 2022-08-18

**Authors:** Maulida Mazaya, Yung-Keun Kwon

**Affiliations:** 1Research Center for Computing, National Research and Innovation Agency (BRIN), Cibinong Science Center, Jl. Raya Jakarta-Bogor KM 46, Cibinong 16911, West Java, Indonesia; 2School of IT Convergence, University of Ulsan, 93 Daehak-ro, Nam-gu, Ulsan 44610, Korea

**Keywords:** pleiotropy, gene–gene interactions, Boolean network dynamics, signaling networks, feedback loops

## Abstract

Pleiotropy, which refers to the ability of different mutations on the same gene to cause different pathological effects in human genetic diseases, is important in understanding system-level biological diseases. Although some biological experiments have been proposed, still little is known about pleiotropy on gene–gene dynamics, since most previous studies have been based on correlation analysis. Therefore, a new perspective is needed to investigate pleiotropy in terms of gene–gene dynamical characteristics. To quantify pleiotropy in terms of network dynamics, we propose a measure called in silico Pleiotropic Scores (sPS), which represents how much a gene is affected against a pair of different types of mutations on a Boolean network model. We found that our model can identify more candidate pleiotropic genes that are not known to be pleiotropic than the experimental database. In addition, we found that many types of functionally important genes tend to have higher sPS values than other genes; in other words, they are more pleiotropic. We investigated the relations of sPS with the structural properties in the signaling network and found that there are highly positive relations to degree, feedback loops, and centrality measures. This implies that the structural characteristics are principles to identify new pleiotropic genes. Finally, we found some biological evidence showing that sPS analysis is relevant to the real pleiotropic data and can be considered a novel candidate for pleiotropic gene research. Taken together, our results can be used to understand the dynamics pleiotropic characteristics in complex biological systems in terms of gene–phenotype relations.

## 1. Introduction

Pleiotropy is the phenomenon in which one gene can result in multiple phenotypes or traits [1,2,3]. In human genetic diseases, it means that different mutations within the same gene cause different pathological effects [4,5]. This becomes an important contributor in identifying a novel function of individual genes with respect to gene–gene interactions [6,7] in system-level biological diseases [8,9]. In this regard, many methods have been modeled to understand the pleiotropy. For example, an experimental study [10] performed several laboratory cultures with the *nath-10 polymorphism* and explained its pleiotropic role in the evolution of a cryptic genetic variation in *C. elegans*. In another study, a statistical analysis [11] using canonical correlation analysis identified a novel candidate pleiotropic associations between genetic variants and phenotypes. In addition, a few computational models [12,13] deployed a pairwise combination of genome-wide association data from the complex disease pleiotropy analysis and modular gene expression analysis. Another study using a metabolic model [14] conducted constraint-based simulations for *E. coli* and *S. cerevisiae* and found that pleiotropy is an emergent property of metabolic network. Finally, protein–protein interaction network analysis was also widely used, and some studies [15,16,17] confirmed pleiotropic effects in biological molecular function, which lead to complex diseases. Despite the interesting observations in previous studies, most of the previous approaches focused on the pleiotropy analysis induced by undirected molecular correlation networks. Therefore, a new approach is needed to investigate the pleiotropy induced by a directed signaling network because it can explain the pleiotropy caused by a gene–gene dynamical relationship.

To quantify the pleiotropic degree of a gene in terms of network dynamics, we proposed in silico pleiotropic score $\left(sPS\right)$, which is a measure to represent how differently a gene affects the dynamics of other genes against different mutations, such as knockout [18,19] and over-expression [20] mutations. In this study, we employed the Boolean network model [21,22] to simulate the network dynamics. A Boolean network model implicitly assumes that all biological components are described by binary values, and their interactions represented by Boolean regulatory functions [23], and it is well-known to capture the silent dynamical properties of biological networks [23,24]. For example, it has been used to analyze oncogene rules in *Non-small cell lung cancer* [25], to model the *C. albicans* yeast for hyphal transition [26], to a matrix cell density sensing to contact inhibition, proliferation, migration, and apoptosis [27], or to illustrate the regulatory effects in cervical cancer [28]. A previous study showed that the dynamics influence of a gene to another genes has some interesting structural characteristics in the signaling network [29]. This study can be extended because the pleiotropy is understood as the difference of the dynamics influence against different mutation types. In this study, through intensive investigations with a signaling network, we observed that most of the dynamically affected genes were related to the experimentally proved pleiotropic genes [30]. Moreover, we found that sPS is negatively correlated with the previous standardized method pleiotropy [2]. Further, we investigated the relationships of sPS with structural properties and found that they have highly positive correlations with degree/in-degree/out-degree, feedback loops, and centrality measures such as closeness, betweenness, stress, and eigenvector, in the signaling network. This implies that the more central of genes, the more pleiotropic. Finally, we found some biological evidence confirming that sPS analysis is relevant to the experimental pleiotropic database, and it can be used for novel candidate pleiotropic gene characteristics. Through a network visualization, we observed that most novel candidate pleiotropic genes are closely located to the known pleiotropic genes. Taken together, these results help to understand the importance of dynamics pleiotropic in complex biological systems in terms of gene–phenotype relations.

## 2. Materials and Methods

### 2.1. Datasets

To examine the in silico dynamics-based pleiotropy, we employed a dataset of a cellular signaling network with 1659 genes and 7964 interactions [31], which was constructed from Kyoto Encyclopedia of Genes and Genomes network (KEGG) database [32]. We then retrieved the list of the pleiotropic genes from human phenotype ontology (HPO) database [30,33], where (a) a gene $v_{i}$ is associated with n phenotypes, and (b) a phenotype p is associated with n genes. In our analysis, this definition was used to sort KEGG genes that are associated with any phenotypes in (a) and (b), and to compute their degree of pleiotropy by the number of phenotypes of those genes. Further, we classified all KEGG genes with respect to the functional importance genes by annotating them with cancer genes, drug-target genes, essential genes, tumor-suppressor genes, oncogenes, and disease genes based on TCGA CBioportal [34,35], DrugBank [35,36], DEG [37], TSGene [38,39], ONGene [40], and DisGenet [41,42] databases, respectively (See Appendix A). We note that this study is not limited to the pleiotropic genes identified from the public databases, because there are other ways to explore the pleiotropy, for example, lab experiments [43] using high-throughput morphometric analysis of hundreds of thousands of single cells in the budding yeast *Saccharomyces cerevisiae* or experimental design in *Drosophila simulans* with genome sequencing [44] to study pleiotropy.

### 2.2. Boolean Network Model

To examine pleiotropic dynamics of genes induced by different types of mutations in a large-scale network, we applied a synchronous Boolean network model [23], which is one of the simplest computational methods to elucidate the network dynamics [22] and has been used to examine complicated behaviors of biological networks [45,46,47]. A Boolean network is represented by a directed graph $G\left(V,A\right)$, where $V=\left\{v_{1},\;v_{2},\;\ldots ,\;v_{N}\right\}$ is a set of nodes, and A⊆V×V is a set of directed links. Each $v_{i}\in V$ has a value of 1 (on) or 0 (off), which means the possible states of the corresponding elements. A directed link $\left(v_{i},v_{j}\right)$ represents a positive (activating) or a negative (inhibiting) relationship from $v_{i}$ to $v_{j}$ ($v_{i}$ and $v_{j}$ are called source and target nodes of the link, respectively). Let $v\left(t\right)$ denote the state of node v at time step t. When a state of $v_{i}$ at time t+1 is determined by the values of $k_{i}$, with other nodes $v_{i_{1}},\;v_{i_{2}},\;\ldots ,v_{i_{k_{i}}}$ with a link to $v_{i}$ at time t, the update rule of $v_{i}$ is represented by a Boolean function $f_{i}:{\left\{0,1\right\}}^{k_{i}}\to \left\{0,1\right\}$. Then, all nodes are synchronously updated, and here, we implemented a nested canalyzing functions (NCFs) model [48,49] to describe an update rule $f_{i}$ as follows:
(1)$$f_{i}\left(v_{i_{1}}\left(t\right),\;v_{i_{2}}\left(t\right),\;\ldots ,v_{i_{k_{i}}}\left(t\right)\right)=\left\{\begin{array}{c} O_{1}\;\;\;\;if\;v_{i_{1}}\left(t\right)=I_{1}\\ O_{2}\;\;\;\;if\;v_{i_{1}}\left(t\right)\neq \;I_{1}\;\mathrm{and}\;v_{i_{2}}\left(t\right)=I_{2}\\ O_{3}\;\;\;\;if\;v_{i_{1}}\left(t\right)\neq \;I_{1}\;\mathrm{and}\;v_{i_{2}}\left(t\right)\;\neq \;I_{2}\;\mathrm{and}\;v_{i_{3}}\left(t\right)=I_{3}\\ \vdots \\ O_{k_{i}}\;\;\;\;if\;v_{i_{1}}\left(t\right)\neq \;I_{1}\;\cdots \;v_{i_{k_{i}-1}}\left(t\right)\;\neq \;I_{k_{i}-1}\;\mathrm{and}\;v_{i_{k_{i}}}\left(t\right)=I_{k_{i}}\\ O_{def}\;\;otherwise \end{array}\right.$$
where $I_{m}$ and $O_{m}\;\left(m=1,\;2,\ldots ,k_{i}\right)$ represent the canalyzing and canalyzed Boolean values, respectively, and $O_{def}$ is generally set to $1-O_{k_{i}}$. In addition, we specified all $I_{m}$ and $O_{m}$ values independently and uniformly at random between 0 and 1. We note that many biological networks were successfully represented by NCFs [50,51,52], and NCFs also properly fit biological experiments’ data [49] including pleiotropy analysis [11]. Those support that NCFs can describe the network dynamics considerably similarly to those real biological networks.

In a Boolean network, a network state at time t can be denoted by a list of state values of all nodes, $v\left(t\right)=\left[v_{1}\left(t\right),\;v_{2}\left(t\right),\;\ldots ,\;v_{N}\left(t\right)\right]\in {\left\{0,1\right\}}^{N}$. Next, every network state transits to another network state through a set of Boolean update functions $F=\left\{f_{1},f_{2},\ldots ,f_{N}\right\}$ and eventually converges to either a fixed point or a limit-cycle attractor starting from its initial state. This attractor represents the diverse biological network behaviors such as homeostasis or oscillation. The definition of the attractor is defined as follows.

**Definition** **1.***Let*$v\left(0\right),\;v\left(1\right),\;\cdots ,$*be a network state trajectory starting at*$v\left(0\right)$*. Then, the attractor denoted by *$G,\;F,v\left(0\right)$*is represented by an ordered finite list of network states *$\left[v\left(\tau \right),v\left(\tau +1\right),\;\ldots ,v\left(\tau +p-1\right)\right]$*where *τ*is the smallest time step such that *$v\left(t\right)=v\left(t+p\right)$*for *∀t≥τ*with*$v\left(i\right)\neq v\left(j\right)\;$ for $\;\forall i\neq j\in \left\{\tau ,\tau +1,\ldots ,\tau +p-1\right\}$*. Herein, *
p
*represents the attractor length.*

In this study, the examination of attractors is needed to find the affected genes. The affected genes were obtained based on our previous work about gene–gene dynamics influence networks [29]. To implement this, we specified a set of initial states,  S, and computed a state trajectory starting at every $v\left(0\right)\in S$ until an attractor is found. We note that the network dynamics can depend on the initial network states.

### 2.3. Computation of In-Silico Pleiotropic Scores $\left(sPS\right)$

Given a gene subject to different types of mutations, we propose sPS of the gene to represent how much the other genes are differentially affected in terms of the dynamics that can be used to deepen the study of existing measures of pleiotropy analysis. Specifically, we considered two mutations, knockout [19,53] and overexpression [54] mutations. The knockout mutation represents the effect of suppressing the expression of a gene or the pharmaceutical inhibition of the secondary messenger production or kinase/phosphate activity [18,55]. On the other hand, the overexpression mutation represents the effect of gene expression change [20,56]. In the Boolean network model, these mutations describe scenarios where the state of mutated gene is frozen to 0 (off) state and 1 (on) state, respectively, during a mutation duration time T. In this study, T is a parameter to denote the mutation duration time, and thus a mutation is effective only for t≤T. This mutation duration time is considered important since it can affect the mutation process of molecular interaction networks [57,58]. Taken together, these mutations can be implemented by changing F into $F^{\prime }$ for t≤T as follows:(2)$$F^{\prime }=\left\{f_{1},\ldots ,f_{i-1},\alpha ,f_{i+1},\ldots ,f_{N}\right\}$$
where α is a set to 0 and 1 in the case of the knockout and the overexpression mutations, respectively. Note that the update rule of $v_{i}$ is restored to $f_{i}$ after the time step T.

To compute sPS of a gene, we employed the notion of the gene–gene dynamics influence used in the previous work [29]. Given a Boolean network $G\left(V,A\right)$ with a set of nodes $V=\left\{v_{1},\;v_{2},\;\ldots ,\;v_{N}\right\}$ specified by a set of corresponding update rules $F=\left\{f_{1},f_{2},\ldots ,f_{N}\right\}$, we generate a set of random initial states S. We first define the dynamics influence of gene $v_{i}$ on gene $v_{j}$, which represents how much the states sequence of $v_{j}$ is changed by a mutation subject to $v_{i}$ as follows (see Appendix A for an illustrative example):
For each initial state $v\left(0\right)\in S,$ we obtain two attractors $G,\;F,v\left(0\right)$ and $G,\;F^{\prime },v\left(0\right)$ in the wild-type and the $v_{i}$-mutant networks, respectively. For convenience, let $G,\;F,v\left(0\right)=\left[v\left(\tau \right),\;v\left(\tau +1\right),\;\ldots ,\;v\left(\tau +p-1\right)\right]$ and $G,\;F^{\prime },v\left(0\right)=\left[v^{\prime }\left(\tau^{\prime }\right),\;v^{\prime }\left(\tau^{\prime }+1\right),\;\ldots ,\;v^{\prime }\left(\tau^{\prime }+p^{\prime }-1\right)\right]$.We compute a distance between $G,\;F,v{\left(0\right)}_{j}$ and $G,\;F^{\prime },v{\left(0\right)}_{j}$ defined as follows:(3)$$d\left(v\left(0\right),v_{i},v_{j}\right)=\underset{m\in \left[0,\;e-1\right]}{\min}\frac{\sum_{l=0}^{c-1}I\left(v_{j}\left(\tau +l+m\right)\neq v_{j}^{\prime }\left(\tau^{\prime }+l\right)\right)}{c}$$
where c and e are the least common multiple and the greatest common divisor, respectively, of p and $p^{\prime }$, and $I\left(condition\right)$ is an indicator function where outputs 1 if condition is true, and 0 otherwise. As a result, $d\left(v\left(0\right),v_{i},v_{j}\right)$ represents the minimum ratio of a bitwise difference between the states sequence of $v_{j}$ in the wild-type and the $v_{i}$-mutant attractors over the least common period (c) of the two attractors.Lastly, we compute the dynamics influence of $v_{i}$ on $v_{j}$ denoted by$\;\mu \left(v_{i},v_{j}\right)$ by averaging out $d\left(v\left(0\right),v_{i},v_{j}\right)$ over a set of initial states in S as follows:(4)$$\mu \left(v_{i},v_{j}\right)=\frac{\sum_{v\left(0\right)\in S}d\left(v\left(0\right),v_{i},v_{j}\right)}{\left|S\right|}$$

Then, let $v_{i}$ an arbitrary source gene in V. Based on the dynamics influence, we can denote a set of affected genes as $\left\{v_{j}\in V|\mu \left(v_{i},v_{j}\right)>0\right\}$. Let $V_{k}$ and $V_{o}$ the sets of affected genes with respect to the knockout and the overexpression mutations, respectively. We then define sPS of a gene $v_{i}$ as follows:(5)$$sPS\left(v_{i}\right)=\frac{1}{\left|S\right|}\underset{v\left(0\right)\in S}{\sum }\left[1-\frac{\left|V_{k}\cap V_{o}\right|}{\left|V_{k}\cup V_{o}\right|}\right].$$

It represents the proportion of the genes that are included in the symmetric difference of $V_{k}$ and $V_{O}$ among their union. Figure 1 shows an illustrative example of computing sPS. Let $v_{1}$ a node subject to the knockout or the overexpression mutation. Through a computation of the dynamics influence from $v_{1}$ to $v_{2}$, $v_{3}$ and $v_{4}$, we obtained the sets of affected genes regarding the knockout and the overexpression mutations, $V_{k}=\left\{v_{2},v_{3},v_{4}\right\}$ and $V_{o}=\left\{v_{2},v_{4}\right\}$, respectively. Thus, $sPS\left(v_{1}\right)=1/3$ which implies that the node $v_{1}$ is pleiotropic because $sPS\left(v_{1}\right)>0$.

### 2.4. A Standardized Measure of Degree of Pleiotropy

A previous research used a standardized pleiotropic measure [2] to compute phenotypic effects in the baker’s yeast *S. cerevisiae*. Given a gene, they examined the average and the standard deviation of the number of transformed traits from wild-type cells, denoted by $m_{wt}$ and SD, respectively, and the number of transformed traits from a cell deficient of the gene, denoted by $m_{d}$. In addition, they defined a standardized measure by the z-transformed pleiotropic score (zPS) as follows:(6)$$zPS=\frac{m_{d}-m_{wt}}{SD}$$

We note that this notion of pleiotropy is different from that in sPS. The former focused on the standardized cardinality of the set of affected genes. To examine the notion of zPS in our work, we set $m_{d}$ to the number of affected genes (i.e., $\left|V_{k}\right|$ or $\left|V_{O}\right|$). In addition, we specified $m_{wt}$ and SD as the average and the standard deviation of the number of associated phenotypes or traits of HPO database for every KEGG gene. In this way, we calculated zPS of the KEGG associated with HPO database and compared with our sPS of them.

### 2.5. Structural Characteristics of Pleiotropic Genes

It is known that the structural characteristics of genes in biological network related to their dynamics stability [59,60]. Here, we considered the following structural properties to investigate the relations to sPS.

A feedback loop (FBL) means a sequence chain of nodes where any node is not repeated except the starting and the end nodes [59,61]. In a given network $G\left(V,E\right)$, an FBL is a closed simple cycle in which all nodes except the starting and ending nodes are not revisited; in other words, a path P of a length $L\left(\geq 1\right)$ is represented by a sequence of ordered nodes $u_{1}\to u_{2}\to \ldots \to u_{L+1}$ with no repeated nodes except $u_{1}$ and $u_{L+1}$. Hence, the P is called a feedback loop if $u_{1}=u_{L+1}$. It was known that FBLs play important roles for controlling dynamics behavior of signaling networks [61,62,63,64].Centrality properties including the closeness defined as the reciprocal of the average distance from a node to every other node [4], the betweenness defined as the ability of a gene to control communication between genes through the shortest paths [65], the stress based on enumeration of the shortest paths [66], and the eigenvector represented by the principal eigenvector of the adjacency matrix of the network, where each node affects all of its neighbors [52].Degree of nodes represents the number of edge upon a gene link to another gene [4,64]. In addition, in-degree and out-degree mean the degree of the incoming and the outgoing links only, respectively.

### 2.6. Random Network Generation

To verify that the results of sPS in the real molecular interaction networks are relevant with randomly structured networks, we generated random networks using the Barabási Albert (BA) model [67], which is a kind of network growth model with a preferential attachment scheme, or called a probabilistic mechanism, where a new node is free to connect to any node in the network, whether it is a hub or just has a single link.

### 2.7. Parallel Computation

For efficient in silico simulations, we implemented the computational program using PANET [46], which is an analysis tool of the network dynamics using the OpenCL library (The recent version is available at http://panet-csc.sourceforge.net/, accessed on 26 November 2019). This allows us to compute large number of attractors in parallel by assigning each initial random state in Equations (3)–(6) to a processing unit of CPUs and/or GPUs.

### 2.8. Statistical Analysis

In this work, we conducted all statistical analysis using the Pearson correlation coefficient and the Student *t*-test using MedCalc Statistical Software version 13.0.6 (MedCalc Software bvba, Ostend, Belgium; http://www.medcalc.org; 2014) [68].

## 3. Results

In this work, we simulated sPS of all genes in the KEGG signaling networks using a Boolean network model (see Section 2). A total of 1000 initial states were randomly generated to calculate sPS, and the mutation duration time *T* was varied from 14 to 20 considering the network size of KEGG $\;(\left|V\right|=1659$).

### 3.1. Comparison of sPS with the Observational Pleiotropy

To show that our approach is relevant to the real phenotype data, we plot a contingency table between the degree of pleiotropy between our in silico model and HPO (Table 1). We first listed every KEGG gene that is associated with any phenotypes in the HPO database (see Appendix A). Next, we specified the degree of pleiotropy with respect to the HPO database as the number of phenotypes or traits of the gene in HPO (‘HPO-associated’). Further, we specified the degree of pleiotropy with respect to our in silico model (the mutation duration time T was set to 20) as the number of KEGG affected genes by the knockout and the over-expression mutations in sPS (‘sPS-associated’).

We then selected top 10 HPO-associated genes that are also related to sPS. As shown in the table, there are some genes that show a high degree of HPO-associated pleiotropy but a low degree of sPS-associated pleiotropy. This shows that our method ‘sPS-associated’ supported the HPO-associated genes and can be used to analyze whose pleiotropic phenomenon is known in terms of network dynamics.

Next, we investigated the relationship between sPS and zPS by varing the mutation duration time (see Figure 2). As shown in Figure 2, there was a negative correlation between them irrespective of the mutation duration time T (All *p*-values < 0.0001 using *t*-test). This implies that our measure is different from the previous measure. This is because zPS focused on the influential extent of the mutation, whereas sPS considers the degree of influential difference caused by a pair of differences. In this regard, sPS can convey a novel viewpoint of pleiotropy from the previous approach.

### 3.2. Relation of sPS and the Functional Importance Genes

Some pleiotropic genes are relevant to many functionally important genes such as cancer genes, drug targets, essential genes, tumor suppressors, oncogenes, and disease genes. For example, it is known that cancer is one of the lead causes of death in human population [69,70], which comes from the accumulation of sequential mutations resulting from cell abnormalities or genetic instability [71,72]. Accordingly, it is not surprising that pleiotropic analysis has become very common in explore different cancer phenotypes [73]. Another examples is the investigation of drug-target genes through network-based analysis [74,75], which shows a significantly different connectivity, more feedback loops, and more evolutionary than non-drug target genes [76]. Thus, drug targets were considered potential cancer therapeutics [77], and recently, they have been recognized as new therapeutic targets for pleiotropic genes [78]. Furthermore, it has been identified that the deletion of any essential genes can lead to death or infertility [79] and tend to be associated with human disease genes [5,80]. Inspired by those results, we investigated the relationship of sPS with functionally important genes. Firstly, we specified every KEGG gene into ‘cancer’, ‘drug-target (DT)’, ‘essential’, ‘tumor-suppressors (TSG)’, ‘oncogene (OCG)’, and ‘disease gene (DG)’ (see Appendix A in detail). Secondly, we examined their sPS values and classified them into two subset groups, ‘non-Zero  sPS’ and ‘Zero-sPS’. We compared the proportion of the functionally important genes between the two groups. Figure 3a–f shows the result of cancer genes, drug targets, essential genes, tumor suppressors, oncogenes, and disease genes, respectively. As shown in the figure, the ratio in the non-zero sPS group is significantly larger than that in the zero-sPS group for all types of functionally important genes and all mutation duration time T. In other words, the functionally important genes tend to be more pleiotropic than the other genes based on our in silico model. It is interesting that this result is relevant with some previous studies. For example, the gene *TERT* was found to be a pleiotropic cancer gene associated with 12 different cancer types [81], while *PTPN2* was confirmed as a pleiotropic gene associated with several autoimmune diseases [82]. This result indicates the promising usefulness of sPS to predict the unknown pleiotropic role of functionally important genes.

### 3.3. Relation of sPS and the Structural Characteristics

Some previous studies have shown that the structural characteristics of a gene in signaling networks are related to its dynamical behavior [59,60]. In this regard, we examined the relation of sPS with the structural characteristics, specifically, the degree, the involvement of feedback loops, and some centrality measures (Figure 4). For every KEGG genes, we first compared the correlation coefficients between degree/in-degree/out-degree of genes with sPS values.

As shown in Figure 4a, all degrees showed significantly positive correlations, irrespective of the mutation duration time T. It means that the sPS values tend to be larger as the number of degree/in-degree/out-degree gets larger. In addition, we observed that the correlation coefficient of sPS with in-degree was relatively lower than those with degree and out-degree. In fact, the difference of the average sPS between two gene groups classified by in-degree values was not as large as that when two gene groups were classified by degree or out-degree values (see Appendix A). It is interesting that degree shows more apparent relation than either specific sub degree type. Next, we examined the relation of sPS with the involvement of feedback loops. Thus, we classified each gene into ‘FBL’ and ‘No FBL’ groups if the gene was involved with any feedback loops or not, respectively, and then compared the average sPS values of the groups. As shown in Figure 4b, the average sPS value of ‘FBL’ group is significantly larger than that of ‘No FBL’ group. This implies that a gene tends to be more pleiotropic when it is involved with feedback loops. Moreover, we computed the correlation coefficient between the number of feedback loops and sPS values and found significant positive relations (see Appendix A; All *p*-values < 0.0001 using *t*-test). This implies that feedback loops can play an important role in pleiotropy, as indicated in a previous study [83]. This result is intriguing because many previous studies have shown the relation of feedback loops with various dynamical behavior of biological networks [62,84]. For example, the FBL plays role in amplifying (positive feedback loop) or inhibiting (negative feedback loop) of the intracellular signals [62,84], related to disease comorbidity [85], or protein–protein interaction [64]. Thus, this result can add the importance of feedback loop structure in terms of the network dynamics. Finally, we examined the relations of centrality measures such as closeness, betweenness, stress, and eigenvector with sPS values and found that all of them have positive correlations (all *p*-values < 0.0001 using *t*-test), irrespective of the mutation duration time (Figure 4c). In other words, it is likely that the more central gene in signaling networks shows a higher sPS value. In particular, the correlation coefficient of closeness, which indicates how closely a gene is located to other genes in a network, was the largest. On the other hand, the correlation coefficient of stress was the lowest. It is interesting that our centrality result is consistent with some previous results showing that pleiotropic genes were more central in protein interaction networks [4,17]. In addition, we examined the correlation coefficient of sPS values with degree/in-degree/out-degree of nodes, feedback loops, and centrality measures in the BA random networks and could observe consistent results (see Appendix A). This implies that our results are principles not only in real networks but also in artificially structured networks.

### 3.4. Biological Evidence of Pleiotropic Genes Based on sPS

To reveal novel candidate pleiotropic dynamics by the sPS measure, we profiled the genes with high sPS values from the KEGG network. A total of 29 genes are shown in Table 2. Among them, ten genes were known to be the real pleiotropic genes, for example, the *PIK3CA* (*PIK3*) gene, which is found to be most pleiotropic among targets of drugs abuse in pharmacological experiments [86], and the *ABL1* gene, which is expressed in all tissues of mice pleiotropic phenotype T-cell signaling [87]. In addition, it was reported that *EGFR* gene is a marker of pleiotropic effects in underlying kidney function and cardiovascular disease [88]. It was clear that those genes were involved with many cancer types, associated with drug targets, essentials, and disease genes. On the other hand, we found 19 novel candidate genes in Table 2 that are not known in the experimental pleiotropic database. This suggests that sPS can predict the unknown pleiotropic genes. As shown in the table, those genes are very interesting because most of them are associated with various functionally important genes such as cancer, drug target, oncogenes, tumor suppressors, essentials, or disease genes. Specifically, all such genes were associated with disease genes. This implies that disease genes tend to be pleiotropic. Further, we map the listed genes in Table 1 into KEGG sub-network (Figure 5; see Appendix A for original network) for visualization. We note that only the listed genes and their neighbors were included in the sub-network. In the figure, the known pleiotropic genes and the novel candidate pleiotropic genes were marked by a blue and a yellow circle, respectively. Interestingly, the novel candidate pleiotropic genes were located closely to the known pleiotropic genes (most of the yellow circle was located from one of blue circle with length 1 in the network). This implies that the novel candidate pleiotropic genes tend to closely interact with the known pleiotropic genes in the signaling network. This was relevant to a study that found that larger effects of pleiotropy can also be caused by correlated effects among traits [89] or the regulatory networks that are so highly interconnected influence neighbor genes to have effects on the core disease genes [90]. In addition, most novel candidate pleiotropic genes were involved in feedback loops in the network. Moreover, their in-degree tends to be smaller than their out-degree. Hence, we can conclude that novel candidate pleiotropic genes are not only associated with certain functional influence genes but also have certain structural characteristics. Taken together, our sPS measure can be more efficiently used to reveal novel candidate pleiotropic genes when it is combined with structural characteristics index.

## 4. Discussion

In this study, we defined pleiotropic scores (sPS), which represents how much the dynamics of a gene is affected against a couple of different mutations, herein the knockout and the overexpression mutations, using a Boolean network model. We investigated our approach using the KEGG signaling network. It was interesting to observe that the affected genes with high sPS values were related to the real pleiotropic data and can explain the dynamical behavior of pleiotropic genes. More interestingly, the various functionally important genes were related to pleiotropic genes. Further, we also found that novel candidate pleiotropic genes tend to closely interact with the known pleiotropic genes in the signaling network. These results will enhance the understanding of dynamical effects on pleiotropic genes, especially in large-scale biological systems. Despite the usefulness of our approach, there are some limitations caused by the Boolean network model used in this study. The first concern is the use of the nested canalyzing function to represent a update of a gene status. However, some previous studies have proven its usefulness in gene regulatory interactions. For example, 133 out of 139 rules compiled from a dataset about a transcriptional regulatory network [50], or 39 out of 42 rules inferred from a dataset about signaling pathways [91] can be classified into the nested canalyzing functions. Another concern is the synchronous update scheme, which is less realistic than the asynchronous update scheme. In fact, it is likely that the genes in the real signaling networks are regulated in an asynchronous update rule. However, it is required to properly specify some unknown parameters to implement the asynchronous scheme. For example, the asynchronous update assumes that only one node can change state at any given moment, and each node has the same probability of being updated [26]. This implies that the asynchronous update is valid only when a correct strategy to choose an update sequence is known. In this regard, a future study will include an approach to infer the update rule from real biological data instead of generating random update rules. In addition, it will extend to a more generalized analysis considering various mutation types.

## 5. Conclusions

Pleiotropy refers to the ability of different mutations within the same gene to cause different pathological effects, and many computational methods have been suggested to unravel the dynamics of the pleiotropy. However, little is known in identifying more complicated dynamical relations of gene pleiotropy, since most of them focused on undirected molecular interaction networks. Therefore, a new perspective is needed to investigate the dynamical characteristics induced by gene–gene pleiotropic. In this work, we proposed a measure to compute gene–gene in silico pleiotropic scores (sPS) representing how much the gene is affected against the different type of mutations on dynamics using a Boolean network model. We considered knockout and overexpression mutations to compute sPS values. Through intensive investigations, we found that some functional importance genes such as cancer, drug-target genes, tumor suppressors, oncogenes, essential genes, and disease genes tend to have non-zero sPS values than other genes. Next, we investigated the relationships of sPS and structural properties and found that there are positive correlations with the number of nodes’ degree/in-degree/out-degree, feedback loops, and centrality measures such as closeness, betweenness, stress, and eigenvector. More interestingly, we were able to find some biological evidence confirming that sPS is relevant to real pleiotropic data and can be used to find novel candidate pleiotropic gene characteristics. Overall, our results suggest the usefulness of sPS in understanding the dynamics pleiotropic in complex biological systems.

## Figures and Tables

**Figure 1 biomolecules-12-01139-f001:**
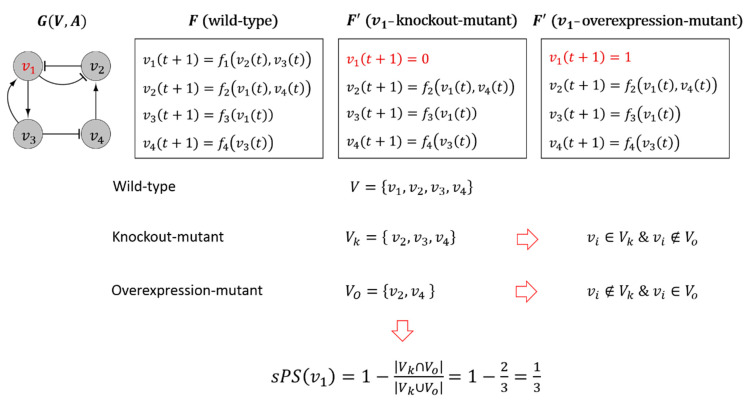
An illustrative example of sPS computation. An example of a signaling network $G\left(V,A\right)$ with a set of update rules F. Let $v_{1}$ a node subject to the knockout or the overexpression mutations. The mutations change F to $F^{\prime }$ where the state value of $v_{1}$ is frozen to 0 and 1, respectively, for t≤T. The sets of genes whose dynamics are influenced by the mutations are identified as $V_{k}$ and $V_{o}$, respectively. Accordingly, sPS of $v_{1}$ is computed as the ratio of the symmetric difference of $V_{k}$ and $V_{o}$ over the union of them.

**Figure 2 biomolecules-12-01139-f002:**
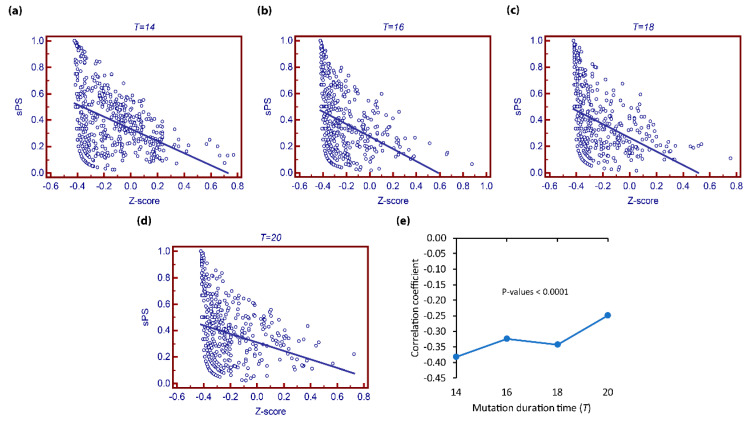
Relationship between sPS and zPS in KEGG network. (**a**–**d**) Relations of sPS and zPS in scatter-plot graph for mutation duration time T=14−20, respectively. (**e**) Correlation coefficients of sPS and zPS. Only genes with positive sPS values were examined. All *p*-values are significant (*p*-value < 0.0001).

**Figure 3 biomolecules-12-01139-f003:**
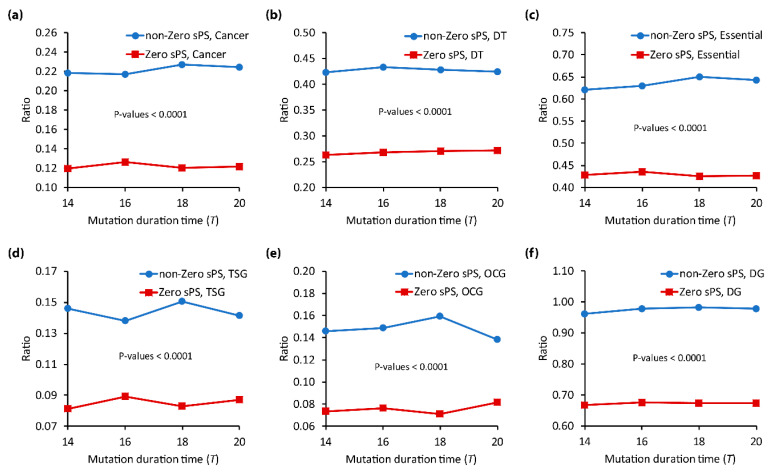
Relation of sPS with the functionally important genes in KEGG network. (**a**–**f**) Results of cancer genes, drug targets, essential genes, tumor suppressors, oncogenes, and disease genes, respectively. In each subfigure, all genes were classified into ‘non-zero sPS’ and ‘zero-sPS’ groups. *Y*-axis is the ratio of the functionally important genes among the total genes in the group. The mutation duration time T was varied from 14 to 20.

**Figure 4 biomolecules-12-01139-f004:**
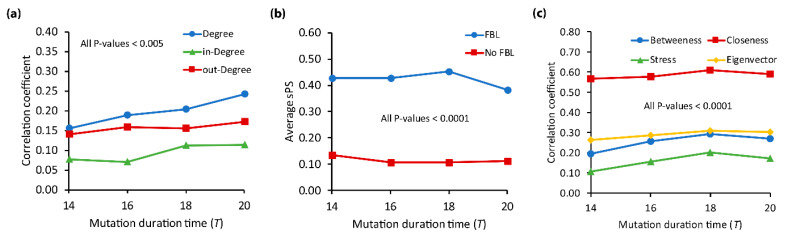
Relation of sPS with structural characteristics in KEGG network. (**a**) Relation to the degree. *Y*-axis values mean the correlation coefficients between sPS and the number of nodes’ degree, in-degree, and out-degree. (**b**) Relation to the involvement of feedback loops. All genes were classified into ‘FBL’ and ‘No FBL’ groups, where a gene involves any feedback loops or not, respectively. *Y*-axis values mean the average of sPS values. (**c**) Relations to the centrality measures such as betweenness, stress, closeness, and eigenvector. *Y*-axis values mean the correlation coefficients between sPS and each centrality measure. Mutation duration time T was set to 14–20.

**Figure 5 biomolecules-12-01139-f005:**
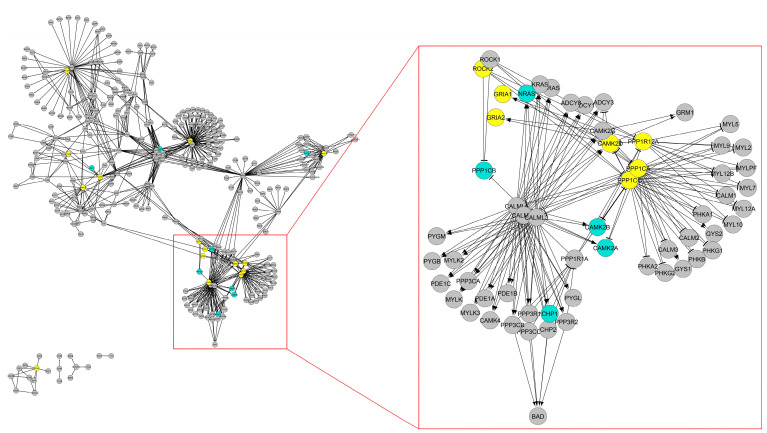
Pleiotropic genes in KEGG sub-network. Arrow-headed and bar-headed lines indicate the activation (positive) and the inhibition (negative) interactions, respectively. The grey circles belong to the non-observed genes. The blue circles represent confirmed pleiotropic genes from the HPO database. The yellow circles represent the novel candidate pleiotropic genes.

**Table 1 biomolecules-12-01139-t001:** The list of genes compared between the degree of pleiotropy by HPO and sPS. The top 10 genes with the highest degree of pleiotropy are chosen by the number of phenotypes in HPO database (‘HPO-associated’) and the number of KEGG genes affected by the knockout and the over-expression mutations in sPS (‘sPS -associated’).

No.	Gene Name	HPO-Associated	sPS-Associated
1	*COL2A1*	842	22
2	*FGFR1*	1045	7
3	*FGFR2*	1137	3
4	*FGFR3*	1006	5
5	*LIMK1*	733	2
6	*NRAS*	728	1
7	*PIK3CA*	751	9
8	*PRKAR1A*	697	1
9	*PTEN*	838	107
10	*TGFBR2*	586	78
Total		8363	235

**Table 2 biomolecules-12-01139-t002:** Significant pleiotropic genes in KEGG network.

No.	Gene Name	HPO	sPS	zPS	NuCancer	DT	ES	TSG	OCG	DG	Deg	In-Deg	Out-Deg	NuFBL
1	*ABL1*	1	0.6	−0.35	23	1	1	0	1	1	27	15	12	4588
2	*PIK3CA*	1	0.7	−0.06	19	1	1	0	1	1	51	45	6	40,101
3	*EGFR*	1	0.45	0.11	21	1	1	0	1	1	73	41	32	225
4	*SERPINA1*	1	0.21	−0.34	60	1	0	0	0	1	1	0	1	0
5	*CAMK2B*	1	0.44	−0.04612	0	1	0	0	0	1	15	10	5	19,612
6	*PPP1CB*	1	0.55	−0.06861	1	0	1	0	0	1	28	3	25	30,526
7	*CAMK2A*	1	0.63	−0.30841	0	1	1	0	0	1	15	10	5	19,612
8	*NRAS*	1	0. 78	−0.40583	1	0	1	0	1	1	44	28	16	20,154
9	*CHP1*	1	0.45	−0.08359	0	1	1	0	0	1	10	8	2	2970
10	*PLA2G6*	1	1	−0.42082	303	1	1	0	0	1	10	10	0	0
11	*IGFBP3*	0	0.45	0.08	1	1	1	1	0	1	1	0	1	0
12	*PRKCA*	0	0.48	0.06	0	1	1	0	1	1	24	7	17	10,054
13	*ITGAM*	0	0.86	−0.4	0	0	0	0	0	1	9	3	6	260
14	*ROCK2*	0	0.59	−0.12	0	1	1	0	0	1	7	3	4	39,206
15	*PPP1CC*	0	0.42	0.09	0	1	1	0	0	1	28	3	28	30,526
16	*PPP1CA*	0	0.52	−0.02	0	1	1	1	0	1	28	3	25	30,526
17	*PRKAA1*	0	0.40	−0.2035	0	1	1	1	0	1	3	0	3	0
18	*PPP1R12A*	0	0.73	−0.27844	0	0	1	0	0	1	15	3	12	7624
19	*CDK2*	0	0.61	−0.36836	0	1	1	1	0	1	10	3	7	4
20	*PPP1CC*	0	0.451327	0.043803	0	1	1	0	0	1	28	3	25	30,526
21	*RALBP1*	0	0.428571	−0.00116	0	0	0	0	0	1	6	2	4	648
22	*CBLB*	0	0.6045	−0.02364	1	0	1	0	1	1	60	4	56	1202
23	*WNT11*	0	0.5455	−0.34588	0	0	1	1	0	1	17	7	10	0
24	*CAMK2D*	0	0.75	−0.27844	0	1	0	0	0	1	15	10	5	19,612
25	*CRK*	0	1	−0.42082	0	0	1	0	1	1	45	31	14	5391
26	*CALML5*	0	0.4455	0.036309	0	0	0	0	0	1	36	9	27	11,451
27	*GRIA1*	0	0.90625	−0.39834	0	1	1	0	0	1	10	9	1	13,272
28	*GRIA2*	0	0.619048	−0.18102	0	1	1	0	0	1	10	9	1	13,272
29	*GNA12*	0	0.317647	0.013827	1	0	1	0	1	1	23	6	17	2528

HPO = 1 means the gene was confirmed in the real observational pleiotropic database, otherwise 0. DT = 1, ES = 1, TSG = 1, OCG = 1, DG = 1 means the gene involves with drug-target, essential, tumor-suppressors, oncogenes, or disease genes, respectively, otherwise 0. NuCancer abbreviates the number of associated cancer types. Deg/In-Deg/Out-Deg denote the values of node degree/in-degree/out-degree. NuFBL abbreviates the number of feedback loops involving gene.

## Data Availability

Data are contained within the article or Appendix A.

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
