# Peer review of "In Silico Pleiotropy Analysis in KEGG Signaling Networks Using a Boolean Network Model"

_biomolecules, 2022, doi:10.3390/biom12081139_

Round 1

Reviewer 1 Report

the submitted manuscript is very interesting, well presented and reports novel data

Any doubts

1. Was the project reviewed and authorized by a research committee? can you put the record All projects, basic, clinical, epidemiological, etc., must be authorized by a committee

2. The manuscript reports a large amount of data, but the discussion section is very small, can you increase it? and discuss their results more precisely?

3. What is the importance of your study? I don't see it described

4 what are your conclusions?

Reviewer 2 Report

This study proposed a measure called in-silico Pleiotropic Scores (???) which is a measure to represent how differently a gene affects the dynamics of other genes against different mutations. The results showed that the model can identify more candidate pleiotropic genes which are not known to be pleiotropic, and these genes tend to have higher ??? values than the other genes. The purpose of the study make sense. However, the experimental design, result section and the language of writing need to be improved throughout. Sentences should be clearer, and more concise. Several suggestions and comments need to be addressed.

1.    Line 29, in the keyword section, “pleiotropy” and “pleiotropic” mean repetition.

2.    Line 236, Table 1. etc. The gene symbol in the manuscript should be Italic, and there are many similar formatting problems, please check them.

3.    In the materials and methods, the importance genes by annotating based on known public databases. So, whether pleiotropy of genes is limited to known public databases.

4.    Line 316, P values should be capitalized and italicized, and there are many similar formatting problems, please check them.

5.    For all Figures, further optimization is required in terms of resolution, layout and fonts.

6.    It is better to provide an R package or other software plug-in, easy to share.

Reviewer 3 Report

Mazaya et al proposed a Boolean network model to study the pleiotropy in terms of the network dynamics. Accordingly, they proposed a measure called in-silico Pleiotropic Scores (sPS) which represents how much a gene is affected against a pair of different types of mutations on a Boolean network model. They found that the model can recognize more candidate pleiotropic genes which are not known to be pleiotropic than the experimental data. Moreover, they uncovered that many types of functionally essential genes tend to have more elevated sPS values than the other genes. They analyzed the relations of sPS with the structural properties in the signaling network and found that there are positive relations to a degree, feedback loops, and centrality measures.  Ultimately, they found some biological evidence indicating that sPS investigation is suitable to the real pleiotropic data and can be viewed for unknown candidate pleiotropic gene search. In the end, these authors suggested that these results can be used to comprehend the dynamics of pleiotropic features in complex biological systems regarding gene-phenotypes relations.

I found this study interesting and relevant in the field.

I found that this study would provide new insight into the knowledge usefulness of sPS to understand the dynamics of pleiotropic in complex biological systems.

The article is well written.

I found the conclusion to be in line with the evidence and arguments presented, and yes, the authors address the main question beautifully.

I have a few suggestions to improve their manuscript.

The authors should provide an introduction to the Boolean network models in the Introduction section, it will be important for the readers. There are some interesting studies available about dynamic Boolean networks such as for (PMID: 35318393) (PMID: 33780439) (PMID: 32913583), authors should cite these studies.

Also, the Boolean network model has some limitations, which can be discussed in the Discussion section.

In subsection 2.5, page number-5 and lines number 201-202 about the Feedback loops, authors should rewrite what means feedback loops and should cite more studies.

Page number-9 lines number-318-319 “This result is intriguing because…..” should cite more studies.
